# The 35th Anniversary of the Discovery of EPR Effect: A New Wave of Nanomedicines for Tumor-Targeted Drug Delivery—Personal Remarks and Future Prospects

**DOI:** 10.3390/jpm11030229

**Published:** 2021-03-22

**Authors:** Hiroshi Maeda

**Affiliations:** 1BioDynamics Research Foundation, Kumamoto 862-0954, Japan; maedabdr@sweet.ocn.ne.jp; 2Department of Microbiology, Kumamoto University School of Medicine, Kumamoto 862-0954, Japan; 3Tohoku University, Sendai 980-8572, Japan; 4Osaka University Medical School, Osaka 565-0871, Japan

**Keywords:** EPR effect, enhanced permeability and retention effect, nanomedicines, cancer therapy, drug delivery, nanotechnology, tumor-selective drug delivery, photodynamic therapy, boron neutron capture therapy

## Abstract

This Special Issue on the enhanced permeability and retention (EPR) effect commemorates the 35th anniversary of its discovery, the original 1986 Matsumura and Maeda finding being published in *Cancer Research* as a new concept in cancer chemotherapy. My review here describes the history and heterogeneity of the EPR effect, which involves defective tumor blood vessels and blood flow. We reported that restoring obstructed tumor blood flow overcomes impaired drug delivery, leading to improved EPR effects. I also discuss gaps between small animal cancers used in experimental models and large clinical cancers in humans, which usually involve heterogeneous EPR effects, vascular abnormalities in multiple necrotic foci, and tumor emboli. Here, I emphasize arterial infusion of oily formulations of nanodrugs into tumor-feeding arteries, which is the most tumor-selective drug delivery method, with tumor/blood ratios of 100-fold. This method is literally the most personalized medicine because arterial infusions differ for each patient, and drug doses infused depend on tumor size and anatomy in each patient. Future developments in EPR effect-based treatment will range from chemotherapy to photodynamic therapy, boron neutron capture therapy, and therapies for free radical diseases. This review focuses on our own work, which stimulated numerous scientists to perform research in nanotechnology and drug delivery systems, thereby spawning a new cancer treatment era.

## 1. Background: Discovery of the Enhanced Permeability and Retention (EPR) Effect, Criticism, and Reality

### 1.1. Status Quo of Enhanced Permeability and Retention (EPR) Effect and Tumor Targeting

Thirty-five years of investigation into the EPR effect [1,2,3,4] have led to the true value of this discovery being increasingly recognized [5,6,7,8]. A recent report by the multinational European Technology Platform on Nanomedicine, set up with the European Commission, stated “*the nanomedicine field is concretely able to design products that overcome critical barriers in conventional medicine in a unique manner*” [9]. This view agrees with the opinions of Lammers et al. [10], Martins et al. [11], and our own [4,5,6,7,12,13]. These viewpoints, however, disagreed with those of Prof. Park [14] and Wilhelm and Tavares [15].

In my opinion, these negative opinions of the EPR effect are based on experimental data for poorly designed nanomedicines. Most of the examples of failed cases reflect the use of so-called nanomedicines with very poor plasma half-lives (t_1/2_) in vivo, or active pharmaceutical ingredients (APIs) in nanomedicines that rapidly became free low-molecular-weight (LMW) drugs. Therefore, similar to the parental LMW APIs, they lacked an essential requirement for nanomedicines of reasonably long t_1/2_ values (i.e., several hours or longer in circulation in vivo). Failures include examples of block copolymer micelle carriers containing doxorubicin such as NK911 (code No. of the drug by Nippon Kayaku Co., Ltd.) or drug-polymer conjugates of inadequate size (less than 30 kDa). Their plasma t_1/2_ values were too short in humans (<3 h). Cases reported by Wilhelm and Tavares [15] demonstrated the same problems. The size of macromolecular drugs that exhibit the EPR effect should be larger than 40 KDa to above 250 KDa, or above a molecular size larger than renal clearance (>5 nm to 100 nm). When the enhancers of the EPR effect are used, it is observed that a limit of this endothelial cell gaps will be increased as discussed later [1,2,3,4,5,6,7]. Furthermore, these findings suggest that the biocompatibility of these conjugates or nanomedicines must not be sufficient to demonstrate good stability during circulation. In contrast, if micellar or liposomal drugs are too stable, they may not release APIs from complexes or nanomedicines, even if they are delivered to tumors via the EPR effect, as is the case with Doxil (doxorubicin [DOX]-containing liposomes), which has a surface coating of polyethylene glycol (PEG) [16].

One should also realize that the milieu into which such drugs are infused is 100% blood, meaning a physiologically acceptable nature is required, and that the drugs are not subject to clearance by reticuloendothelial or phagocytic cells. Blood is quite different from physiological saline or deionized water because it contains many dye-binding proteins; dense negative charges also exist on vascular surfaces and will interact with APIs; and APIs may be abstracted from the micellar complex with APIs (drugs) before getting to tumor. Our previous reviews documented these problems related to failed cases [4,5,6,7,12,13].

### 1.2. Issue Concerns Passive Targeting to Tumor vs. the EPR Effect Driven Tumor Targeting

I want to emphasize, in this occasion, a critical difference between “passive targeting” and “EPR-effect based tumor targeted drug delivery”. During the arterial angiography, a LMW x-ray contrast agent such as Angioconray^®^ is infused intraarterially (i.a.), then this x-ray contrast agent of a LMW nature is taken up more selectively into the tumor tissues than normal tissues. This is indeed passive targeting. However, this LMW contrast agent administrated will be rapidly washed out within a minute or so, as seen by x-ray imaging. In contrast, when macromolecular agents of >40 KDa or albumin binding dye Evans blue is injected i.v., they will be more selectively accumulated in the tumor tissue than normal tissues, and retained in the tumor for a prolonged period, more than several hours to weeks. This does not happen for LMW agents as they will be washed out rapidly. Similar to macromolecular drugs, when the lipidic contrast agent Lipiodol^®^, which is iodinated and ethylated poppy seed oil, is injected into the tumor feeding artery, Lipiodol becomes microparticles as it is broken up during its passing through the branched capillaries. Consequently, Lipiodol behaves like nanoparticles, and it will be retained in tumor tissue selectively more than several hours to months as easily seen by x-ray CAT scan, but this does not happen in normal tissues. This account is discussed in detail later.

Arterial infusion of LMW anticancer agents was tried extensively in the past as well as bolus intratumor injection, but both modalities were not so effective. Then, slow continuous arterial infusion using a infusion pump was conducted, though the drugs being infused will diffuse back quickly to the circulating blood, and result is more or less similar to i.v. infusion.

In conclusion, passive targeting only showed a short period of tumor retention, which is almost insignificant compared to the prolonged time of drug retention seen in EPR driven tumor delivery of macromolecular anti-cancer agents. The key issue here is that the passive targeting of drug does not implicate prolonged tumor retention of the drugs. This is the rationale of the EPR effect driven cancer therapy with longer retention time in tumors, but it needs to use nanomedicines, but not by LMW drugs.

In addition, the initial finding of the EPR effect was based primarily on experiments with tumor models in mice, whereas many large advanced tumors that are frequently seen in clinical situations differ from the small tumors in mice [12,13]. Nevertheless, we have ample evidence of the EPR effect occurring in human cancers. For example, neocarzinostatin (NCS) conjugated to poly(styrene-co-maleic acid) (SMA)—the conjugate named SMANCS—dissolved in Lipiodol^®^ and given intra-arterially accumulated selectively in human solid tumors, as described below. Traditional radioscintigraphy with radioactive ^67^Ga, which binds to the plasma protein transferrin (90 kDa), showed selective accumulation of ^67^Ga in tumors by virtue of the EPR effect. More recently, intravenously injected nanomedicines demonstrated a tumor-selective EPR effect in breast cancer [17] and renal cancer [18].

### 1.3. Inflammation and EPR Effect Observed in Bacterial Infection Protease and Permeability Inducing Factors; Bradykinin and Other Mediators

As a historical aside, before I describe the vascular permeability of solid tumors, I should mention that we first studied bacterial infection and inflammation with a focus on the role of proteases produced by bacteria [19,20,21,22]. We then found that the bradykinin-generating cascade of endogenous proteases was activated by exogenous proteases produced by bacterial infection. That is, the sequence of the cascade was Hageman factor or factor XII → kallikrein → kininogen → bradykinin (kinin) (Figure 1). Kinin is a nonapeptide (RPPGFSPFR) cleaved off from kininogen in the plasma, and it induces vascular permeability, severe pain, and various signaling molecules. All microbial proteases including fungi trigger the cascade system, of which multiple steps are affected (Appendix A) [21,22]. Activated endogenous proteases function in two important pathways: (i) thrombin activation and then fibrin formation, and (ii) kinin generation (Appendix A), which is a key factor in vascular permeability in tumors, bacterial infections, and inflammation [23,24,25,26,27]. Ascitic and pleural effusions in carcinomatosis also largely depend on kinin generation in vivo [23,24,25,26,27,28,29,30].

In large advanced tumors, blood vessels are often occluded or embolized, although individual tumor pathology varies. For example, some liver metastases, pancreatic tumors, and prostate cancers have avascular areas with less vascular density, whereas primary liver and kidney cancers have extremely high vascular densities and therefore a correspondingly significant EPR effect (see also the discussion below). However, animal research ethics committees at most institutions restrict the use of large tumors, more than 5000 mm^3^, in experimental settings. Such large tumors have occluded or embolized tumor blood vessels, as above-mentioned, and the degree of vascular density can be demonstrated by means of arterial angiography with a contrast agent such as Lipiodol^®^.

Since 1983, we have been studying blood vessels and their characteristics [6,7,8,31,32,33,34,35,36] in human cancers of the liver, kidney, lung, and other solid tumors. Contrast-enhanced arterial angiography showed highly stained areas that indeed corresponded to the EPR effect. We also demonstrated the effect of EPR effect enhancers in the above tumors including angiotensin II induced high blood pressure [34,35,36]. In contrast, pancreatic, prostate, and metastatic liver cancers showed low-density staining, thus indicating poor blood flow or avascular nature of tumors. These tumors have either occluded blood flow, or a weak or heterogeneous EPR effect.

## 2. Nanomedicines: Proceeding from Tissue EPR Effects to Tumor Cellular Uptake to Molecular Targets in Tumor Cells

After a nanomedicine has reached a tumor, the drug (the API) must enter the tumor cells and then affect target molecules in the cells. Doxil is delivered to tumor tissues because of its high stability in vivo and does have an EPR effect [16,37], but it has a low rate of API (DOX) release from the liposomes. DOX, liberated from Doxil, also has a low rate of internalization by tumor cells, which is a crucial issue. Although once DOX is internalized into cells, then to the nucleus, it forms an intercalated complex with target DNA. In this case, DOX is retained in the nucleus for a long time [38]. However, slow cell uptake of DOX is more critical before bending to the target. For instance, we found great different rates of internalization of DOX vs. pirarubicin, which is a derivative of DOX in which one mole of extra tetrahydropyranyl group is added. Pirarubicin showed a 10- to 100-fold higher rate of internalization, even though both DOX and pirarubicin possess the same anthracycline structure as their biologically active component [12,39,40] (see Figure 2). This rapid intracellular uptake of free pirarubicin continued even after the conjugation of pirarubicin with the *N*-(2-hydroxypropyl) methacrylamide (HPMA) polymer (Figure 2A). In contrast, the same DOX-polymer conjugate showed an extremely poor cellular uptake, and its biological activity was also poor (Figure 2B). The superior cellular uptake of pirarubicin may be attributed to the pyranyl group (i.e., its structure), which is similar to that of glucose (pyranose). Pyranose can be utilized in the cell uptake step by the glucose transporter system of tumor cells, which is highly upregulated in tumor cells.

With regard to the physicofchemical properties of macromolecular drugs (nanomedicines), we have described the importance of hydrophobicity and pH in the tumor microenvironment, which affects protonation and deionization of the carboxyl group in SMANCS, for instance [41,42,43]. That is, in addition to the styrene group’s hydrophobicity, which results in an affinity to cell membranes; the maleyl carboxyl group becomes a pH sensor in the tumor environment. When the pH becomes lower than neutral, that is, the COO^−^ is fully ionized to the protonated form (–COOH), hydrophobicity increases [41,42,43]. The result is a 100- to 200-fold increase in uptake by tumor cells in culture. As an additional advantage of this amphiphilic polymer conjugate, SMANCS and its parental proteinaceous antitumor agent (NCS) are active against drug (DOX)-resistant cell lines [44].

## 3. Future Prospects for the EPR Effect: Toward Clinical Application

### 3.1. Restoration of Tumor Blood Flow and Augmentation of the EPR Effect

The discussion above on the EPR effect for cancer-selective drug delivery is based on the assumption that tumor blood flow is normal—without vascular embolization, semi-necrotic areas that have poor blood flow, or necrotic tissue with blocked blood flow. However, the EPR effect, as just discussed, is often reduced in clinical settings, which is a most critical issue for proper tumor drug delivery [12,44,45,46]. The success of cancer chemotherapy with nanomedicines as based on the EPR effect thus requires normal tumor blood flow. For this purpose, we have worked on vasodilators or EPR effect enhancers including nitroglycerin [4,5,6,7,8,13,34,35,36], isosorbide dinitrate, L-arginine, and angiotensin I-converting enzyme inhibitors such as enalapril, among others. Our earlier and recent publications have emphasized this topic [8,9,12,13,34,35,36,45,46,47]. In this Special Issue, readers will find other tactics to enhance the EPR effect such as using bubble liposomes, microwaves, and heat [48,49].

In my opinion, very few nanomedicines are available for cancer chemotherapy that fulfill all the ideal requirements for use in patients, although many candidate nanodrugs are under development [11]. Our prototype polymeric drug, for example, the poly(hydroxypropylacrylamide) conjugate of pirarubicin (P-THP), so far seems to meet these requirements, although it needs approval by a regulatory agency before clinical use [47,48,49,50,51]. Many patients who received P-THP as compassionate use in a hospice mostly with stage IV or terminal disease, showed no apparent toxicity at the therapeutic dose level and responded very well to the treatment. Metastatic bone tumors or tumor nodules in the pleural compartment disappeared as expected ([52,53], and unpublished data]).

### 3.2. Arterial Infusion of Nanomedicines with Extremely High Accumulation in Tumors

Another option exists for enhanced tumor-targeted drug delivery. This method has not been so widely used because x-ray angiography and arterial infusion using a catheter requires qualified skills. The method involves application of a lipid formulation of lipophilic nanodrugs and trans-arterial infusion into tumor-feeding arteries via a catheter under x-ray monitoring. This modality produces by far the best tumor-targeted drug delivery as well as tumor imaging [31,32,33,34,35,36] and a tumor/blood ratio of more than 100 can easily be achieved [34,54,55]. We have successfully utilized this technique with SMANCS dissolved in Lipiodol^®^, and the method was approved for clinical use by the Ministry of Health, Labor, and Welfare of Japan. SMANCS in Lipiodol^®^ solution becomes microparticles as it is pushed into arterial vessels, that is, SMANCS/Lipiodol^®^ selectively extravasates into tumor tissues as microparticles, with results that are based on the EPR effect [31,32,33,34,35,36].

Arterial infusion of lipophilic drugs dissolved in Lipiodol^®^ can be so selectively targeted to a tumor that the dose of the drug used in the infusion can be far reduced compared with the conventional systemic (i.e., intravenous) dosage. Therefore, we proposed that the doses for such arterial injections should be based on tumor size, not the body surface area or body weight of a patient [56]. Additionally, infusions for particular tumors such as bronchial, lung, or colon require special attention because a targeted area may suffer damage caused by a high concentration of drug and complications may ensue. For this reason, the dose of the drug should be 1/10 of the liver or gallbladder cancer. It is thus not strictly based on the tumor size [36,56]; high drug concentrations in such tissues with neighboring void spaces may cause the tissue to rupture, the results being perforation and bleeding.

## 4. Enhancement of Cancer Chemotherapy, Utilization of Photodynamic Therapy (PDT), Innovation in Boron Neutron Capture Therapy (BNCT), and Use of Reactive Oxygen Species (ROS)/Reactive Nitrogen Species (RNS) as Scavengers for Cancer and Inflammation via Nanodrugs

### 4.1. Enhancement of Photodynamic Therapy (PDT)

We and others have reported the many advantages of the EPR effect, primarily for cancer chemotherapy with nanomedicines. However, the usefulness of nanomedicines for photodynamic therapy (PDT) and boron neutron capture therapy (BNCT), which have been known for more than a century and several decades, respectively, would be far greater with nanotechnology when LMW photosensitizers (PSs) as well as boron containing drugs were converted to nanomedicine.

With regard to PSs, one can clearly demonstrate tumor-selective accumulation of polymer-conjugated PSs via in vivo models (Figure 3). We developed polymer conjugates of PEG, SMA, and HPMA to LMW zinc protoporphyrin (ZnPP) [4,5,6,13,50,57,58,59,60,61,62] (Figure 3). The PSs yielded fluorescence values above 500 nm and generated singlet oxygen or ROS, which can kill tumor cells. Selective fluorescence can be clearly detected in tumors in vivo (Figure 4A,B). This evidence is clear proof of the EPR effect.

Despite the long history of PDT-use in cancer therapy, its clinical impact has been insignificant. The reasons for this are: (i) most PSs developed so far such as Photofrin and Laserphyrin are of LMW with little EPR effect; and (ii) PSs being used contain a porphyrin chromophore, which is best excited at about 390–450 nm. However, in the past, most human applications used a HeNe laser that emits light only at 633 nm, which is far from the proper excitation wavelength of about 400 nm. Another criticism concerns hemoglobin interference: PSs composed of porphyrin derivatives with excitation wavelengths of about 400 nm will be affected in vivo by hemoglobin, which exists in massive amounts in the blood and will absorb excitation energy that is similar to the wavelength of the PSs being used, so the irradiating light will be absorbed before reaching the PSs. We can assume that the irradiating light will not effectively excite the PSs, which is a consequence of using improper wavelengths (633 nm) to excite PSs. However, this assumption may be true only in heme-rich organs such as the liver, spleen, and blood vessels. In contrast, tumor tissues do not have many blood cells. Red blood cells have a diameter of about 6 µm and cannot easily extravasate into tumor tissues or normal tissues. In addition, some PSs such as HPMA-polymer ZnPP and PEG-conjugated ZnPP have a compact micellar form, so that aromatic rings of the PSs molecules are packed within a close distance of each other. Thus, π–π interactions will quench the fluorescence and no singlet oxygen will be generated (Figure 3B). These PSs will fluoresce after the micelles traverse via endocytosis through cell membranes, which contains the lipid-bilayer into tumor cells and then the micelles disintegrate due to the detergent effect of the lipid bilayer (Figure 3B, in cell, right).

The therapeutic effect depends on both the PS (polymeric PS) dose and the intensity of the irradiating light (Figure 5B,C). We adapted the light source used for conventional endoscopy for this purpose (Figure 4A).

Drawbacks associated with current conventional PDT will not be seen with nano PSs because of the highly tumor-selective nature of the fluorescent nanoprobe, polymer-conjugated protoporphyrin (P-ZnPP) (Figure 4B,C). One problem involves hyper-sensitivity to light: patients who have undergone injections of conventional PSs are required to stay in a dark environment for a few weeks because of hypersensitivity of the skin: PSs will spread throughout the body including normal tissues, particularly the skin of the face and hands.

Our ZnPP has another beneficial effect. Even without light irradiation, it inhibits heme oxygenase (HO-1) as well as heat shock protein-32, and it downregulates oncogene expression [63,64,65]. HO-1 generates carbon monoxide and biliverdin/bilirubin as products of heme degradation by heme oxidation. Both carbon monoxide and bilirubin are potent antioxidants and block the actions of ROS/RNS, which are generated to produce a tumoricidal effect by host macrophages and neutrophils as part of the innate immunity mechanism. Therefore, PEG-ZnPP and SMA-ZnPP have antitumor effects themselves by potentiating tumor cell killing by ROS/RNS that are generated by leukocytes [62,66].

### 4.2. A Hot Progress in Boron Neutron Capture Therapy (BNCT) with Boron Nanomedicines

BNCT, like PDT, has been poorly developed. BNCT utilizes compounds containing ^10^B and thermal neutron irradiation generated by a nuclear reactor or an accelerator. In this modality, ^10^B compounds, as in PDT, must reach the local tumor for the best therapeutic effect without adverse effects. This requirement of tumor selective localization of ^10^B means that the possibility exists for application of EPR effect-based ^10^B-containing nanomedicines. In contrast to radiotherapy with x-ray or γ-ray irradiation, which require oxygen that will become effector ROS molecules, the thermal neutrons of BNCT, however, do not need oxygen molecules. The thermal neutrons need to hit ^10^B atoms, the result being a yield of α-particles and lithium atoms as active principles that can kill cancer cells within a radius of 10 micron (see Figure 6 and Figure 7A′). Current conventional BNCT in clinical settings uses an LMW ^10^B derivative such as boronophenylalanine (BPA). Similar to the situation with chemotherapy with LMW cytotoxic drugs, BPA is not expected to be tumor selective (Figure 7B′). A continuous intravenous infusion of BPA during neutron irradiation is necessary to maintain an adequate boron concentration in the tumor tissue because its urinary excretion is quite rapid.

As Figure 7B illustrates, BPA exists in both normal and cancer tissues. Therefore, BPA may affect normal tissue such as the skin as well as cancer-neighboring normal tissues other than tumor tissue. For instance, when treating an oral cancer with BNCT, vocal cords and superficial skin may be harmed. Use of BNCT thus carries the probability of adverse effects. However, we can avoid this problem by using macromolecular boron derivatives [67].

We recently published a report on such macromolecular boron derivatives in which SMA was first linked with glucosamine (SG) [67]. Glucosamine forms a stable complex with boric acid (SGB complex). Natural boric acid contains about 25–30% of ^10^B, with the remainder being ^11^B. ^10^B-enriched boron derivatives are available, however. The SGB complex forms micelles of about 12 nm, as seen with election microscopy, about 65 kDa in solution (Sephacryl S200), and it can bind with albumin in solution, so that its size increases to more than 120 kDa [67]. This size is ideal for the EPR effect to operate. In experiments with a tumor-bearing mouse model, the accumulation of the SGB complex in tumors was about 10-fold higher than that in all normal tissues including the liver and kidney [67].

The SGB complex has multiple actions in addition to the generation of α-particles such as the inhibition of glycolysis; see reference [67] for details, and Figure 6. Similar to glucosamine, one can conjugate BPA to the SMA polymer, and similar results will be expected, but neither inhibition of glycolysis (suppression of lactic acid formation), nor damage to mitochondria are expected. Preliminary data for neutron irradiation in vitro and in vivo were validated: tumors shrunk without any effects on skin or on toxicity in the liver and kidney, or on blood counts. I can thus envision new possibilities for BNCT with boron nanomedicines, where a new wave is coming.

## 5. Development of ROS and RNS Generators or Scavengers Utilizing the Advantages of Nanodrugs, and Future Clinical Applications

### 5.1. Elimination of Toxic Free Radical ROS/RNS in Infection and Cancer by Using Nanomedicines

Oxygen free radicals, or ROS, and RNS cause various diseases. ROS and RNS species are produced primarily at sites of infection, inflammation, and cancer. Maeda et al. demonstrated that excessive generation of ROS and RNS, together with nitric oxide (NO), occurs during influenza virus infection in mice. These species are responsible for the pathogenesis of influenza and influenza-related pneumonia; they are also associated with other microbial infection, and they also further accelerate viral mutations [68,69,70,71].

We have investigated the effects of a free radical-scavenging enzyme, superoxide dismutase (SOD; MW about 20 kDa), in influenza virus-infected mice. Intravenously injected native SOD was not effective by itself, because the t_1/2_ of native SOD is too short (<1 h), as discussed above. Conjugating SOD to pyran copolymer (pyran-SOD) considerably improved the pharmacological and therapeutic effects, and diseased mice were cured. Namely, mice that received injections of pyran-SOD had a 95% cure, whereas native SOD had no effect on the survival of the mice [68,69].

ROS have no single source, but are initially derived from macrophages or neutrophils, followed by activation of xanthine dehydrogenase to xanthine oxidase (XO) in diseased tissue such as the lung [68,69,70]. In contrast, more extensive production NO is derived from the inducible form of NO synthase in macrophages in the inflamed tissue or in cancer. Two of these molecular species, O_2_^−^ and NO, react quite rapidly in situ and form peroxynitrite, which is more reactive than O_2_^−^ and NO and has highly oxidative and nitrative effects on DNA, RNA, proteins, and lipids. A free radical storm (i.e., NO, O_2_^−^, HClO, ONOO^−^, etc.) are likely operating behind the scenes in this complicated current COVID-19 pandemic and must be controlled [69,70,71,72]. This pandemic may be out of control until we have effective vaccines or antiviral agents as well as control of the ROS/RNS storm [72]. As with ROS, O_2_^−^ is converted to H_2_O_2_ (a less reactive ROS) by SOD, and when myeloperoxidase in neutrophils is accessible to H_2_O_2_ and chlorine, HClO (hypochlorite) will be formed, which will also damage DNA, RNA, proteins, and lipids as well as bacteria, tumors, and normal tissues, the consequence being a triggering of many diseases. ROS/RNS generation thus formed in microbial infection will result in the accelerated formation of mutation unless the formation of ROS/RNS is controlled [73,74,75,76].

Shashni and Nagasaki prepared a unique polymer conjugate of 4-amino-TEMPO, a redox-cycling nitroxide (4-hydroxy-TEMPO; (4-hydroxy-2,2,6,6-tetramethylpiperidine-1-oxyl)-TEMPO), another free radical scavenger with poor pharmacokinetic properties by itself [77]. When they conjugated this redox-sensitive prosthetic group (amino-TEMPO) to a diblock copolymer (PEG) plus [poly(tetramethyl-piperidine-1-oxyl)aminomethylstyrene], the polymer conjugate was superior, with far better pharmacokinetics and showed suppressive effects on tumor growth (see [77]). The finding of this polymer conjugate may be applied to ROS/RNS-related diseases with inflammation or complicated infections such as COVID-19.

### 5.2. Using ROS/RNS Generation to Kill Cancers by Means of ROS-Generating Polymer-Conjugated Enzymes, or Rescuing ROS-Caused Damage by Means of Enzyme Replacement Therapy via Conjugation with Synthetic Polymers

An important early use of PEGylated enzymes was enzyme replacement therapy. Use of PEGylated adenosine deaminase (ADA) for congenital disease is well documented [78]; the t_1/10_ in humans was about one month, which may be better than that for the infusion of recombinant lymphocytes with ADA being the t_1/2_ of normal lymphocytes in general is about a month. Additionally, PEG-L-asparaginase has long been used in clinical situations for patients with leukemia [79]. Its t_1/2_ was 3 min and converted to 56 h, and its t_1/10_ was >11 days. In this context, the HPMA-polymer conjugate of protein may be preferable to PEGylated enzymes because it is so far free from immunogenicity or less immunogenic compared with PEGylated enzymes. Namely, PEGylation generates an anti-PEG antibody, which becomes a problem a few weeks later after initial infusion, even in the case of PEG-L-asparaginase. On the basis of a similar principle, we addressed hyper-bilirubinemia (jaundice). High concentration of bilirubin in blood causes jaundice and at higher concentrations, it becomes toxic to many cells. We PEGylated bilirubin oxidase produced by fungus and found that its t_1/10_ became much higher (1.8 min → 48 h in rats) [80].

We also investigated an opposite direction to utilize ROS generation by XO as a possible cancer cure [81,82]. PEGylated XO (PEG-XO) produced significant antitumor activity after three PEG-XO injections in two weeks; each PEG-XO injection was followed by daily injections of its substrate, hypoxanthine. Here again, native XO alone followed by infusion with hypoxanthine resulted in no therapeutic effect, but conjugation of biocompatible PEG improved the pharmacokinetics of XO and exhibited an EPR effect, and therapeutic benefit was improved.

We later applied a similar strategy to D-amino acid oxidase (DAO), which is another ROS (H_2_O_2_)-generating enzyme. When we injected a D-amino acid such as D-proline or D-alanine to tumor bearing mice i.v., PEGylated-DAO (PEG-DAO) generated H_2_O_2_ in the tumors because of selective tumor accumulation of PEG-DAO by virtue of the EPR effect; this antitumor strategy worked well to control tumor growth in the mouse [83,84]. In a different investigation, Fang et al. achieved successful therapeutic results with polymer (SMA)-conjugated AHPP (4-amino-6-hydroxypyrazolo[3,4-*d*]pyrimidine), an XO inhibitor [85] with an anti-inflammatory and antihypersensitivity activity.

More examples may exist of which I am not aware, but so far, no drugs that utilize free radical generation or scavengers are in clinical use.

H_2_O_2_ generation is an important event in healthy organisms and is essential in that it occurs (predominantly) via NADPH oxidase or other enzymes in leukocytes. Congenital deficiency of NADPH oxidase results in chronic granulomatous disease (CGD), particularly in infant and children because of the lack of H_2_O_2_ or O_2_^−^ to kill bacteria, and constant or chronic infections will lead to CGD. We therefore prepared PEGylated DAO to deliver PEG-DAO to inflamed sites and thus supply ROS, in parallel with administration of D-proline or D-alanine, the DAO substrates. When H_2_O_2_ is generated, it will be converted to the more powerful bactericidal molecule. HClO is generated by neutrophils in the presence of both myeloperoxidase and chloride ion, which will kill bacteria [82,83]. Normal healthy cells contain enzymes for defense against ROS, which is catalase for H_2_O_2_ and SOD for O_2_^−^.

Many cancer cells lack these anti-oxystress enzymes or have downregulated levels of these enzymes, so they are vulnerable to oxystress. Many advanced cancer cells propagate well under anaerobic conditions, and antioxidant enzymes may be lost [6,7,12,81,82] due to elevated levels of hypoxia due to embolization or clotting in the blood vessels [46,86]. To dissolve fibrin clots to activate plasminogen to plasmin, Mei et al. used redox sensitive polymer conjugate, and made enhanced vascular permeability by newly generated plasmin [87], and also modulate an extra cellular tumor environment [86,88]. Thus far, delivery of ROS-generating or scavenging enzymes conjugated to synthetic polymers may be an intriguing therapeutic strategy.

## 6. Concluding Remarks

This Special Issue commemorates my 35th year after discovery of the EPR effect [1,2], and therefore this review includes many of my own papers related to this area. I have focused primarily on synthetic and artificial nanomedicines, so I have not included antibody-linked drugs, cytokines such as interferon, interleukin-6, and tumor necrosis factor-β, and liposomes.

The ultimate purpose of personalized medicine is to provide the best benefits for individual patients. The EPR effect is a ubiquitous phenomenon found in almost all solid tumors, with sizes from less than 1 mm to larger than 10 cm; this effect also occurs in inflamed tissues and applies to biocompatible macromolecules. To utilize the EPR effect or the related drug delivery system more effectively, vascular blood flow must be restored and maintained. Nanomedicines are of prime importance for receiving the benefits of the EPR effect. The issues of vascular flow in tumor tissues is a relatively recent issue in cancer therapy [4,7,12,13,45,47,63,86,88], although vascular embolism in cardiology, for example, has been investigated often for some time, but not much in relation to cancer [13,45,67,86,89].

Various advantages of the unique properties of nanomedicines as well as selective drug targeting to tumors and inflamed tissues were easily demonstrated via pharmacokinetics and pharmacodynamics and imaging; the use of EPR effect enhancers exhibit fewer adverse effects and improved therapeutic results are thus expected when combined with nanomedicines compared with conventional medicines in the future. Nanomedicine is therefore worthy of study and challenges for the benefit of patients. Wider applications of PDT and BNCT as well as strategies to control the ubiquitous undesirable molecules like ROS/RNS are future lines of study (e.g., [77]). The growing knowledge of the tumor microenvironment, as discussed by Subrahmanyam and Ghandehari in this volume [86], will provide many clues for the future delivery of nanomedicines and may make use of many intelligent or sophisticated sensors or probes.

## Figures and Tables

**Figure 1 jpm-11-00229-f001:**
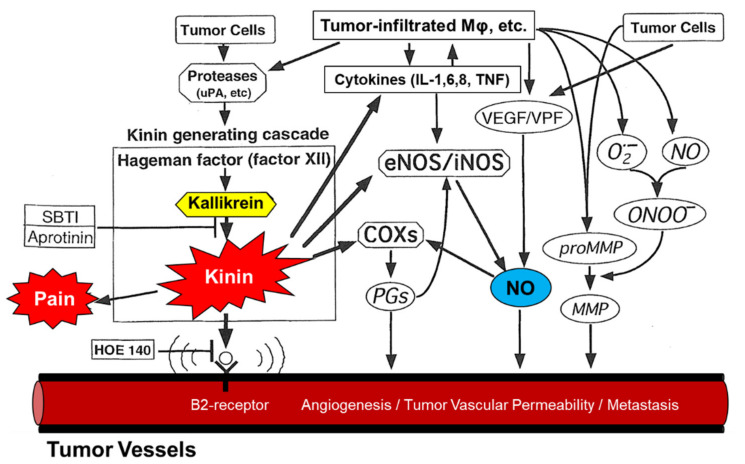
The enhanced permeability and retention (EPR) effect in tumor vasculature. The mechanism of this tumor-selective macromolecular drug targeting depends on various effectors affecting vascular tone, as shown here. Aprotinin is an inhibitor of kallikrein; HOE-140 is a peptide antagonist of kinin. SBTI, soybean trypsin inhibitor; NO, nitric oxide; eNOS, endothelial nitric oxide synthase; iNOS, inducible form of nitric oxide synthase; COXs, cyclooxygenases; PGs, prostaglandins; MMP, metalloproteinase; ONOO^−^, peroxynitrite; O_2_^−^, superoxide anion radical; MΦ, macrophage; VEGF, vascular endothelial growth factor; VPF, vascular permeability factor; uPA, urokinase plasminogen activator; IL, interleukin; TNF, tumor necrosis factor; B2 receptor, bradykinin B2 receptor (see also Appendix A, adapted from ref [23]).

**Figure 2 jpm-11-00229-f002:**
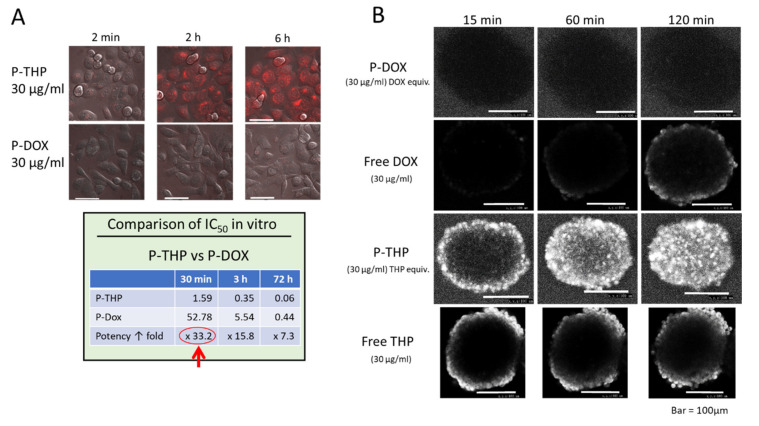
Comparison of the cellular uptake of P-THP—the poly(hydroxypropylmethaacrylamide [HPMA]) conjugate of pirarubicin (THP)—and P-DOX (HPMA polymer-DOX conjugate) by human pancreatic cancer cells (SUIT-2) in vitro. (**A**) Polymeric P-THP shows a far greater uptake by tumor cells compared with P-DOX: at 30 min, P-THP had a 33.2-fold higher uptake, and its cytotoxicity had greatly increased (see Table at lower left). (**B**) Penetration of P-DOX, DOX, P-THP, and THP into spheroidal tumor colon cancer (Adapted with permission from ref. [39,40]. 2016 American Chemical Society, 2019 American Chemical Society). Far greater penetration of P-THP into the tumor spheroid (similar to Figure 2, Table) is seen.

**Figure 3 jpm-11-00229-f003:**
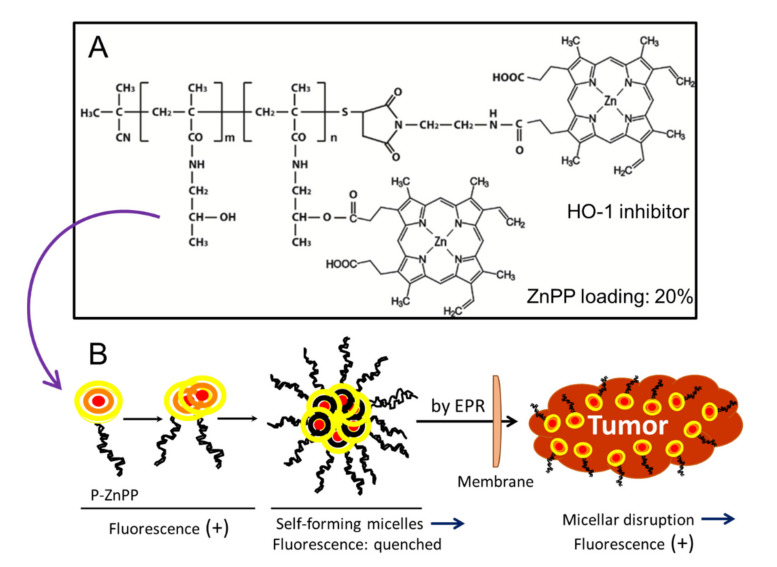
Self-assembling PS polymer conjugates of HPMA and ZnPP. (**A**) Chemical structure of the HPMA-PS polymer conjugate. (**B**) Polymer-ZnPP in solution. Spontaneous micelles were formed. Quenching occurs in the self-forming micellar form of P-ZnPP, which leads to a lack of fluorescence in the micellar form. When tumor cells take up these micelles, the micelles disintegrate during the traversing lipid bilayer due to its amphiphilic nature. Then, fluorescence becomes positive and singlet oxygen (ROS) are generated in the tumor upon light irradiation (**B**). ZnPP itself also inhibits heme oxygenase-1 (HO-1) and suppresses tumors (see text for details).

**Figure 4 jpm-11-00229-f004:**
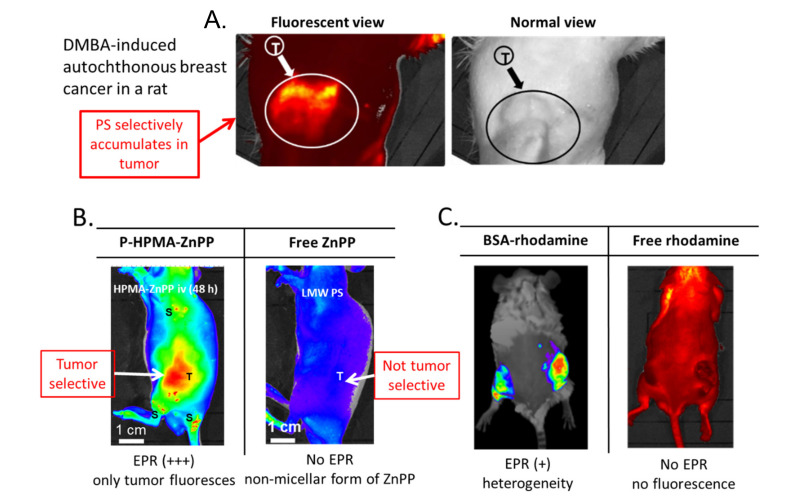
Fluorescence imaging of breast cancer in a rat and of implanted S180 tumor in a mouse, after intravenous injection of P-ZnPP. (**A**) DMBA (7,12-dimethylbenz[*a*]anthracene)-induced breast cancer in a rat. Under fluorescent light (left) and under normal light (right). (**B**) Fluorescent image of nano-PSs: polymeric HPMA-ZnPP (P-HPMA-ZnPP) and free ZnPP. (**C**) Rhodamine-conjugated bovine albumin (BSA) vs. free rhodamine. Images show no accumulation of LMW free PSs in tumors (**B**,**C**). (**A**, adapted from [58]; **B**,**C**, adapted from ref. [4]).

**Figure 5 jpm-11-00229-f005:**
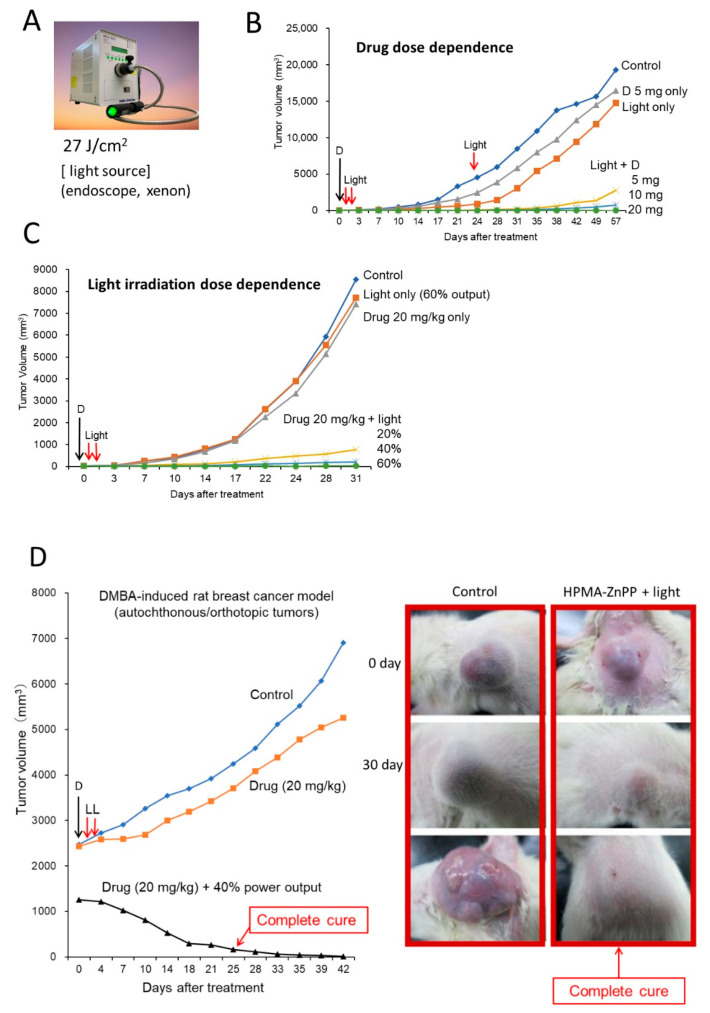
Photodynamic therapy (PDT) with polymeric PSs. (**A**) View of the light source for the endoscope; a xenon lamp was used. (**B**) Dose dependence of P-ZnPP dosage, marked D. (**C**) Dose of light irradiation intensity. The D indicates the time of drug injection of P-ZnPP in B and C. The power of irradiation light (%) is relative to full power output of the endoscope (100%). (**D**) Results of PDT treatment of DMBA-induced breast cancer in rats. L, light irradiation. D, drug injection. Control received only light. Boxed images at right show growth and suppression of tumor after PDT and P-ZnPP treatment (**right**) and tumor without treatment (**left**).

**Figure 6 jpm-11-00229-f006:**
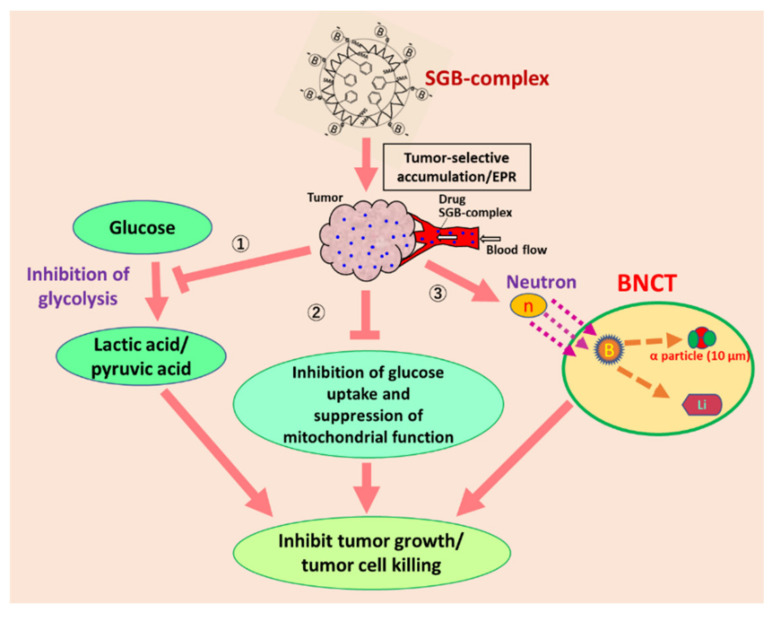
This represents the mode of action of poly(styrene-co-maleic acid) conjugated glucosamine (SGB-complex), which forms complex with boric acid, then forms micelles (~15 nm) and exhibits the EPR effect, about 10 times more boron accumulation in the tumor than other normal tissue [67]. When this SGB-complex is used, it exhibits three different cell killing mechanisms as denoted by “①, ②, and ③” in this figure. By neutron irradiation at right, ③, it elicits the production of α-particles which will kill the tumor cells within 10 micron radius. SGB-complex is rapidly incorporated into the tumor cells and inhibit both glycolysis ① and production of lactic acid; ② it also affects the structural integrity of mitochondria, and its size will shrink and suppress ATP production in the cells (Reprinted with permission from ref. [67]. 2020 Elsevier Ltd.).

**Figure 7 jpm-11-00229-f007:**
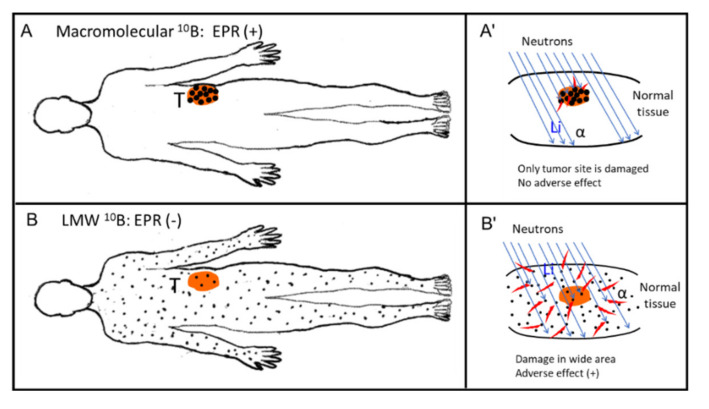
Body distribution of boron-containing drugs. (**A**) Body distribution of a macromolecular ^10^B compound (e.g., SGB-complex). (**B**) Distribution of an LMW ^10^B compound. In (**A**), boron-containing micelles such as the SGB-complex accumulates predominantly in tumor tissue (T), with the accumulation being about 10 times greater than that of a LMW compound or all other normal tissues in (**B**). (**A’**,**B’**) at right represent enlarged views of the neutron irradiation sites. In (**A’**), only tumor tissue is damaged: boron micelles (back dots) are evident only in the tumor (T). In (**B’**), neighboring normal tissue to tumor the boron compound are distributed in most normal tissues such as skin, which will be then be damaged. Red specks around black dots indicate the area of emission of α-particles. (**B’**) shows that a wide area of tissue is damaged in (**B’**) (adverse effect).

## Data Availability

The data that support the findings in this article are available from the corresponding author, H.M. upon reasonable request.

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
