# Peer review of "The 35th Anniversary of the Discovery of EPR Effect: A New Wave of Nanomedicines for Tumor-Targeted Drug Delivery—Personal Remarks and Future Prospects"

_jpm, 2021, doi:10.3390/jpm11030229_

Round 1

Reviewer 1 Report

This manuscript summarized progression of the Enhanced Permeability and Retention (EPR) effect concept over 35 years after its discovery by the author of this manuscript.  The manuscript is well written and covers all the aspects. 

Author Response

Thank you for your comment.  No specific reply needed to describe.

Reviewer 2 Report

The article 'The EPR Effect 35 Years after Its Discovery —— New Waves of 2
Nanomedicines for Tumor-Targeted Drug Delivery: Personal 3
Remarks and Future Prospects' is a very important article and the author have nicely summarized the evidence of EPR and its importance. 

Below are few recommendations of minor nature. 
1- The author should describe the optimal size range of nanoparticles/drugs required for EPR effect. The literature describes a very wide range of particle sizes (from 100 nm to 800 nm) and therefore, the effect of particle size should be discussed in this paper for further clarification. 

2- Although its not discussed in the article, but there is a debate of whether EPR (or passive targeting) is more effective as compared to active targeting mechanisms such as ultrasound mediated destruction of drug loaded nanobubbles reported by Exner group doi.org/10.1016/j.xphs.2019.05.004.

It would be beneficial for the reader if the author explains that whether or not active targeting could also get benefits from EPR. 

3- Various researchers have reported drug loaded nanoparticles and their cellular uptake and discussed the EPR effect. It would be optimal if the author include more references like  https://doi.org/10.3390/cancers11101464 ,  10.1158/1078-0432.CCR-05-0343 and  https://doi.org/10.3390/cancers11071024 among several others. 

Author Response

[Q 1] About the molecular size of nanomedicine applicable for EPR driven tumor targeting.

[Reply 1]

(1) Defective blood vessel of individual tumor is so variable that there is no narrow range of molecular size threshold to define.  One thing for certain is apparent size of the nanomedicine is above 40 KDa in globular proteins, it could be applicable to even more than 500 KDa. (Now added this notion in p. 2).

(2) Because of the amount of vasodilating factors such as NO, bradykinin, etc, are so variable, the gaps between each endothelial cells are variable, which makes critical gap distance for extravasation (permeability can be defined).  It is a dynamic phenomena, as we described in many reviews (eg. Fnag J et al, ADDR 2020, 157, 142-160, W. Islam et al., Mol. Cancer Ther. 2018, see new discussion of “(2) Issue concerns passive targeting to tumor vs EPR effect driven tumor targeting” in p. 2).

[Q 2] Regarding “the passive targeting” vs “EPR effect driven tumor targeting”.

[Reply 2] 

There is a confusion, in that, EPR effect is equal to passive targeting.  However, there is a great difference in that the concept of “passive targeting”, which does not contain the concept of retention time.  Passive targeting can last for less than a minute as one can demonstrate with LMW X-ray contrast agent (imaging agent) as seen by arterial angiography.

   For this purpose a new Section “(2) Issue concerns passive targeting to tumor vs EPR effect driven tumor targeting” is added in p.2, after line 67.

[Q 3] Cellular uptake and EPR effect. 

[Reply 3]

Cellular uptake is only possible that nanodrugs are accessible to cancer cells in order to be taken up into cells.  That is the reason that EPR effect is the first step before reaching to the targets in cells in which nanodrugs should be best suited to get accessible to cancer cells. To make that possible, the EPR driven access to tumor tissue is needed.  Those references suggested by the referees are now included in appropriate places. Reference of nuclear DNA as drug binding sink of in the cell (by T. Allen’s) is added a